# PCSK9 Confers Inflammatory Properties to Extracellular Vesicles Released by Vascular Smooth Muscle Cells

**DOI:** 10.3390/ijms232113065

**Published:** 2022-10-28

**Authors:** Maria Francesca Greco, Alessandra Stefania Rizzuto, Marta Zarà, Marco Cafora, Chiara Favero, Giulia Solazzo, Ilaria Giusti, Maria Pia Adorni, Francesca Zimetti, Vincenza Dolo, Cristina Banfi, Nicola Ferri, Cesare R. Sirtori, Alberto Corsini, Silvia Stella Barbieri, Anna Pistocchi, Valentina Bollati, Chiara Macchi, Massimiliano Ruscica

**Affiliations:** 1Department of Pharmacological and Biomolecular Sciences, Università degli Studi di Milano, 20133 Milan, Italy; 2Centro Cardiologico Monzino, Istituti di Ricovero e Cura a Carattere Scientifico (IRCCS), 20133 Milan, Italy; 3Department of Clinical Sciences and Community Health, Università degli Studi di Milano, 20133 Milan, Italy; 4Department of Life, Health and Environmental Sciences, Università degli Studi dell’Aquila, 67100 L’Aquila, Italy; 5Unit of Neuroscience, Department of Medicine and Surgery, Università degli Studi di Parma, 43124 Parma, Italy; 6Department of Food and Drug, Università degli Studi di Parma, 43124 Parma, Italy; 7Department of Medicine, Università degli Studi di Padova, 35100 Padua, Italy; 8Department of Medical Biotechnology and Translational, Università degli Studi di Milano, L.I.T.A., 20133 Milan, Italy

**Keywords:** extracellular vesicles, PCSK9, atherosclerosis, inflammation, vascular smooth muscle cells

## Abstract

Vascular smooth muscle cells (VSMCs) are key participants in both early- and late-stage atherosclerosis and influence neighbouring cells possibly by means of bioactive molecules, some of which are packed into extracellular vesicles (EVs). Proprotein convertase subtilisin/kexin type 9 (PCSK9) is expressed and secreted by VSMCs. This study aimed to unravel the role of PCSK9 on VSMCs-derived EVs in terms of content and functionality. EVs were isolated from human VSMCs overexpressing human PCSK9 (VSMC^PCSK9^-EVs) and tested on endothelial cells, monocytes, macrophages and in a model of zebrafish embryos. Compared to EVs released from wild-type VSMCs, VSMC^PCSK9^-EVs caused a rise in the expression of adhesion molecules in endothelial cells and of pro-inflammatory cytokines in monocytes. These acquired an increased migratory capacity, a reduced oxidative phosphorylation and secreted proteins involved in immune response and immune effector processes. Concerning macrophages, VSMC^PCSK9^-EVs enhanced inflammatory milieu and uptake of oxidized low-density lipoproteins, whereas the migratory capacity was reduced. When injected into zebrafish embryos, VSMC^PCSK9^-EVs favoured the recruitment of macrophages toward the site of injection. The results of the present study provide evidence that PCSK9 plays an inflammatory role by means of EVs, at least by those derived from smooth muscle cells of vascular origin.

## 1. Introduction

Atherosclerosis is a chronic inflammatory disease characterized by the formation of lipid-laden plaques in the arterial wall. Atherosclerotic lesions are characterized by a lifetime accumulation and transformation of lipids, inflammatory cells, vascular smooth muscle cells (VSMCs) and necrotic cell debris in the intimal space underneath a monolayer of endothelial cells [1,2]. During lesion growth, VSMCs switch from a contractile to a proliferative state and migrate into the intima [3]. Phenotypically modified VSMCs can influence neighbouring cells in a paracrine manner by secreting a wide range of bioactive molecules, some of which are packed into extracellular vesicles (EVs) to be delivered to recipient cells favouring atherosclerotic processes [4,5].

The release of membrane-bound vesicles, namely, EVs, is a universally conserved cellular process that occurs in all three domains of life (i.e., Archaea, Bacteria and Eukarya) [6]. EVs deliver their cargo (e.g., proteins, lipids and nucleic acids) among cells as a form of intercellular communication [7,8]. Thus, EVs represent a route of autocrine/paracrine/endocrine communication between different cell types and even distant organs [9]. Besides regulating normal cell homeostasis, in the context of cardiovascular diseases, EVs are considered key players in the initiation and progression of atherosclerosis [10], with pro- and anti-atherothrombotic properties depending on their molecular cargo, cell of origin, and the stimulus triggering the release [11].

In said context, the impact of proprotein convertase subtilisin/kexin type 9 (PCSK9) cannot be underestimated. An association between PCSK9 and the release of EVs derived from atherosclerotic components (i.e., platelets, endothelium, monocytes/macrophages and neutrophils) was recently described [12]. Besides being one of the main regulators of low-density lipoprotein receptor (LDLR), the hypothesis that PCSK9 is directly involved in atherogenesis is well supported by observations showing the expression of PCSK9 in human atherosclerotic plaques (abundant in the “shoulder” regions of vulnerable atherosclerotic plaques), in endothelial cells, macrophages and VSMCs [13,14]. In particular, VSMCs release significant amounts of PCSK9 [15] (more than endothelial cells) with maximal release at low shear stress, with a greater PCSK9 expression in aortic branch-points and aorta-iliac bifurcation of the mouse aorta that express low shear stress [16]. Conversely, data pertaining to the impact of human PSCK9 on VSMCs (e.g., whether PCSK9 inhibits or favours cell proliferation) are contrasting [17,18].

Based on these premises and gaps in the knowledge concerning the paracrine role of PCSK9 in the feed-forward-loop of the atherosclerotic process, the present study is aimed at dissecting whether PCSK9 influences the phenotypic composition of EVs released from human VSMCs and how these EVs impact the cell-to-cell communication among components of atheromatous lesions. To pursue this goal, a model of human VSMCs stably overexpressing PCSK9 was generated and the related EVs (VSMC^PCSK9^-EVs) used in functional experiments by using both in vitro models of endothelial cells, monocytes and macrophages and in vivo embryos of zebrafish (*Danio rerio*). Collectively, the current study demonstrates that when VSMCs express a higher amount of PCSK9, the released EVs carry a pro-inflammatory phenotype.

## 2. Results

### 2.1. Characterization of VSMCs Overexpressing PCSK9

The first step was to assess whether VSMCs transfected with a vector carrying human PCSK9 efficiently overexpressed PCSK9 compared to cells transfected with a mock plasmid. Since PCSK9 was FLAG-tagged (Asp-Tyr-Lys-Asp-Asp-Asp-Asp-Lys), by WB analysis we detected the presence of this short sequence (Figure 1A). A higher expression of PCSK9 in VSMCs^PCSK9^ compared to VSMCs^WT^ was also confirmed at gene and protein levels (Figure 1B and Table 1) as well as in the cell culture medium (Figure 1C). Since our group previously demonstrated that VSMCs freshly isolated from *Pcsk9^−/−^* mice have a slower proliferation rate compared to those isolated from their counterpart (*Pcsk9^+/+^*) [17], we investigated if the overexpression of PCSK9 in human VSMCs could have affected this feature. Cell count analysis showed that the proliferation rate of human VSMCs^PCSK9^ was faster than that of control cells (VSMCs^WT^) with a stepwise increment at 24, 48 and 72 h (Figure 1D). Based on this evidence, we also tested a possible switch towards a synthetic phenotype. VSMCs^PCSK9^ displayed a more rounded shape, while VSMCs^WT^ exhibited an elongated and spindle-shape phenotype (Figure 1E). Gene expression of actin alpha 2 (Acta2) and calponin was significantly decreased (Figure 1F). These findings were further corroborated by gene ontology (GO) enrichment analysis conducted on the 33 proteins which were highly expressed in VSMCs^PCSK9^ compared to VSMCs^WT^ (Table 1 and Appendix A). A significant enrichment (FDR < 0.05) was found in the categories pertaining to cell differentiation and regulation of cell morphogenesis involved in differentiation (Figure 1G and Appendix A).

### 2.2. Characterization of VSMC^PCSK9^-Derived EVs

Since we harvested EVs from VSMCs cultured in medium without serum, to avoid the risk of contaminating EVs with intracellular or apoptotic vesicles, cells were stained with FITC annexin V and PI. They did not reveal any significant difference in the percentage of live cells between those cultured for 24 h in 10% FBS or only medium 0.1% BSA (Figure 2A). The next step was to assess if a time-dependent release of EVs happened. By means of a time-course experiment, we set 24 h as the best time point to collect the highest amount of EVs (Appendix A). To reassure that the overexpression of PCSK9 in the donor cells (VSMCs) could not have affected concentration, size and shape of EVs, an Nanosight Tracking Analysis (NTA) analysis was performed. There were no differences in the concentration and size between EVs released by VSMCs^PCSK9^ (from now on, VSMC^PCSK9^-EVs) and their counterpart VSMC^WT^ (from now on, VSMC^WT^-EVs). After normalization for cell count, mean concentrations were 926.17 ± 815.26/mL/cell count and 625.17 ± 235.23/mL/cell count, respectively, whereas mean sizes were 233.16 ± 16.3 nm and 235.78 ± 29.78 nm, respectively. We further analysed the effect of the overexpression of PCSK9 in terms of distribution of vesicle concentrations for each EV size. In Figure 2B, the mean concentration for all EV sizes (i.e., from 30 to 700 nm) is depicted. The comparisons among the EV sizes are shown in Figure 2C, where the two *p*-values obtained, comparing VSMC^WT^-EVs vs VSMC^PCSK9^-EVs (negative binomial linear regression models at each size) showed no significant differences. No changes were found also in the case of ultrastructural morphology (Figure 2D). TEM analysis showed the presence of typical rounded, whole and undamaged large EVs and small EVs in both cell lines. The integrity of EVs was confirmed by the presence of an unbroken bilayer membrane, visible as a thin white filament surrounding the electron-dense EV content (Figure 2D). Furthermore, VSMC^PCSK9^-EVs and VSMC^WT^-EVs were positive for tetraspanins CD9 and CD63, and for β1-integrin and Alix (Figure 2E,F). Finally, a representative immunoblotting analysis showed that VSMC^PCSK9^-EVs carried a higher amount of PCSK9 compared to VSMC^WT^-EVs (Figure 2F and Table 2).

Specifically, untargeted proteomic analysis identified 14 proteins significantly more abundant in VSMC^PCSK9^-EVs (Table 2 and Appendix A).

These data were further analysed by using STRING (v 11.5) to identify enriched GO terms and create the networks. The GO analysis reported a significant enrichment (FDR < 0.05) in the following categories: lipoprotein extracellular matrix structural constituent, lipoprotein particle receptor binding, very-low-density lipoprotein particle receptor binding and signalling receptor binding (Figure 3A and Appendix A). Concerning EV-miRNA content, compared to those isolated from VSMC^WT^-EVs, VSMC^PCSK9^-EVs differently expressed 6 miRNAs: hsa-miR-34c, hsa-miR-29, hsa-miR-148b, hsa-miR-221 and hsa-miR-125b were downregulated; hsa-miR-49 was upregulated (Figure 3B). Thus, the genes targeted by these miRNAs were compared with genes associated with atherosclerosis (n = 20,445) and inflammation (n = 467) downloaded from DisGeNET v 7.0. The Venn diagram shows that 54 genes were associated with atherosclerosis and inflammation (Figure 3C and Appendix A).

### 2.3. Impact of EVs on Pro-Inflammatory Markers in Endothelial Cells, Monocytes and Monocyte-Derived Macrophages

Before assessing the impact of VSMC-derived EVs on recipient cells, the effectiveness of EVs uptake was tested. EVs isolated from donor cells (both VSMCs^WT^ and VSMCs^PCSK9^) were labelled with PKH67, a specific cell membrane dye, and then 5 × 10^8^ EVs/mL were transferred for 24 h to the recipient cells, THP-1 monocytes. The range of 10^8^ EVs/mL is one of the most used in the literature [19,20]. The fluorescence of THP-1 exposed to labelled EVs was almost 5-fold higher compared to THP-1 receiving EVs not labelled with PKH67 (Figure 4A; Appendix A). We initially tested the expression of adhesion molecules and pro-inflammatory markers in EA.hy926 endothelial cells upon exposure to EVs. A significant rise in gene expression of interleukin-6 (*IL-6*), vascular cell adhesion molecule 1 (*VCAM-1*), endothelin 1 (*ET-1*), intercellular adhesion molecule 1 (*ICAM-1*) and E-selectin was found in EA.hy926 cells exposed to VSMC^PCSK9^-EVs compared to those exposed to VSMC^WT^-EVs (Figure 4B). In line with these findings, 24-h-exposure of THP-1 and THP-1-derived macrophages to VSMC^PCSK9^-EVs dramatically increased the gene expression of *CCL2* (MCP-1/chemokine (C-C motif) ligand 2), *IL-1α*, *IL-1β*, *IL-6* and *IL-8* (Figure 4C,D). The raised inflammatory milieu driven by VSMC^PCSK9^-EVs was confirmed by the protein expression of phosphorylated form of STAT3 (signal transducer and activator of transcription 3) and SOCS3 (suppressor of cytokine signaling-3), two well-known cell signalling pathways involved in atherosclerosis [21]. In both THP-1 monocytes and THP-1-derived macrophages, VSMC^PCSK9^-EVs compared to VSMC^WT^-EVs sharply raised the phosphorylation levels of STAT3 with a subsequent downregulation of SOCS3 (Figure 4E,F). To strengthen these findings, a label-free mass spectrometry-based approach was applied to analyse the secretome of THP-1 exposed to EVs (Appendix A). To avoid any contamination from EVs, the secretome was collected after ultracentrifugation at 110,000× *g* for 4 h at 4 °C. The subsequent GO analysis showed that, compared to VSMC^WT^-EVs, VSMC^PCSK9^-EVs led to an enrichment in pathways involved in immune response, immune effector process, response to stress and cellular response to cytokines (Figure 4G and Appendix A).

### 2.4. Effect of EVs on Cell Migration, ox-LDL Uptake and Mitochondrial Functionality in Recipient Cells

A further step of our experimental design was to evaluate the impact of VSMC^PCSK9^-EVs on the migratory capacity of monocytes and macrophages. Compared to VSMC^WT^-EVs, VSMC^PCSK9^-EVs favoured the migration of recipient cells towards a gradient of 10 ng/mL CCL2 [22]. After 10 h, 6.3% of THP-1 exposed to VSMC^PCSK9^-EVs, compared to 3.5% of those exposed to VSMC^WT^-EVs, migrated from the upper part of the transwell support to the bottom of the chamber (Figure 5A). Concerning macrophages, to overcome technical issues encountered with migratory assay and derived-THP-1-macrophages, J774 macrophages were used [23]. VSMC^PCSK9^-EVs decreased the migratory capacity by 54% compared to J774 exposed to VSMC^WT^-EVs (Figure 5B). All-in-all, these findings are in line with the evidence showing that during plaque formation, there is an increase migration of monocytes while macrophages show a reduced egress [1]. As a further step aiming to evaluate the proatherogenic phenotype carried by EVs released from VSMC^PCSK9^, we found a 22% rise in the uptake of oxLDL-derived cholesterol by THP-1-derived macrophages (Figure 5C). To explain such a phenomenon, the protein expression of scavenger receptor CD36 was evaluated. We found that the protein expression of CD36 was modestly, but significantly, upregulated in response to VSMCs^PCSK9^-EVs (Figure 5D). Since accumulating evidence confirms the involvement of mitochondria in atherosclerosis [24], the direct measurement of the oxygen consumption rate was tested in THP-1 recipient cells. VSMC^PCSK9^-EVs significantly reduced basal and maximal respiration by 17% and 45%, respectively, compared to VSMC^WT^-EVs (Figure 5E). In line with this evidence, we found that VSMCs^PCSK9^-EVs increased the percentage of glycolytic proton efflux rate (PER) (+6%) with a decrement (−17%) in the ratio between mitochondrial oxygen consumption rate (mitoOCR) and glycolytic proton efflux rate (glycoPER) (Figure 5F). Collectively, the drop in the oxidative phosphorylation and the rise in glycolytic activity portend to an inflamed cellular phenotype.

### 2.5. Evaluation of Macrophage Recruitment upon EVs Injection in Zebrafish

Zebrafish represents a very useful in vivo experimental model to study changes in the inflammation milieu [25]. To generate a systemic delivery, EVs were microinjected into the duct of Cuvier of 48 hpf embryos. Twenty hours post-injection (hpi), VSMC^PCSK9^-EVs, compared to VSMC^WT^-EVs, elicited a significant increase in the gene expression of IL-1β and IL-8 (Figure 6A). To better elucidate the contribution of macrophages as mediators of the pro-inflammatory state, the effect of VSMC^PCSK9^-EVs on the macrophage migratory capacity was assessed. Embryos at 3 dpf were subjected to local injection in the close cavity of the hindbrain ventricle (Figure 6B,C) and intramuscularly (Appendix A) [26]. In both cases, the injection of VSMCs^PCSK9^-EVs increased the recruitment of macrophages toward the local injection site at 6 hpi in comparison to the same amount of VSMC^WT^-EVs (>27–30% macrophage recruitment on average), as observed through l-plastin immunostaining assay (Figure 6C and Appendix A). A representative fluorescence overlaid with brightfield images of trunk (lateral view) of 3 dpf embryos treated with CFSE-stained EVs (green) is shown in Appendix A.

## 3. Discussion

In the intertwined relationship among cellular components involved in the molecular biology of atherogenesis, the results of the present study pointed out the cholesterol-independent role played by PCSK9 in atheroma formation. By influencing the content of EVs released from VSMCs, PCSK9 favours a pro-inflammatory phenotype in in vitro models of endothelial cells, monocytes and macrophages as well as in zebrafish embryos. Considering that VSMCs play critical roles in multiple vascular diseases, including atherosclerosis [27], the rationale of using VSMCs as a proxy is based on the following assumptions: VSMCs express significant levels of PCSK9 [18,28] which are able to downregulate the expression of LDLR expression in macrophages [15] and monocytes [29]; a raised expression of PCSK9 is found in VSMCs under pro-inflammatory stimuli [29] and in regions at low shear stress (3–6 dyn/cm^2^) [16]; VSMCs isolated from *Pcsk9*^−/−^ mice, compared to those of *Pcsk9*^+/+^ mice, have a reduced proliferation rate and a suppressed migratory capacity [17,30]; PCSK9 mediates a switch towards a pro-calcific phenotype of VSMCs [5]; there is an increased number of the synthetic phenotype of VSMCs that express PCSK9 in acute aortic dissection samples [31]; in a synthetic or proliferative non-contractile state, VSMCs exhibit an increased production of EVs [32].

EVs originate from diverse subcellular compartments and are released into the extracellular space to be engaged in cellular crosstalk via the transfer of selective biomolecular cargo (mostly comprising lipids, proteins and nucleic acids) to recipient cells in an autocrine, paracrine or endocrine manner to regulate cell function [33]. The effects of EVs on target cells are highly influenced by the parental cell, stimuli, local environment and, especially, composition. EVs appear to have a multifaceted role and depending on their cargo, they can either facilitate or hamper the development of atherosclerotic lesions [34]. With these concepts in mind, before characterizing the content of EVs, we first noted that PCSK9 favoured a phenotypic switch of human VSMCs towards a synthetic phenotype. This evidence is in line with previous observations reporting that PCSK9 induces cell proliferation in mouse- or rat-derived smooth muscles [30,35]. However, it must be acknowledged that one study, by using both human aortic VSMCs and mouse VSMCs knockout of *Pcsk9*, demonstrated the opposite, namely, PCSK9 inhibits proliferation and induces polyploidization, senescence and apoptosis [18].

The second step consisted in the phenotypical characterization of EVs. The shape, dimension and integrity were not affected by the overexpression of PCSK9. However, according to changes in the phenotype of VSMCs, VSMCs^PCSK9^-EVs carried a different cargo of proteins (n = 14) and miRNAs (n = 6) compared to VSMC^WT^-EVs. GO analysis found an enrichment in extracellular matrix structural constituents. The extracellular matrix is involved in all aspects of vascular pathobiology. This network regulates the biomechanical properties of blood vessels and the phenotype of the cells that reside in them, such as endothelial cells, VSMCs, adventitial fibroblasts and infiltrating immune cells from the circulation. Though many cell types mentioned can synthesize extracellular matrix macromolecules, VSMCs are considered the most prominent ones [36]. Regarding PCSK9, although this protein could potentially have effects on atherosclerosis that are independent of cholesterol levels, the direct mechanism of involvement is still unknown, driving a gap in the knowledge of such a predominant player in cardiovascular disease [37]. In line with this evidence, VSMCs^PCSK9^-EVs-miRNAs targeted 54 genes which were associated with both atherosclerosis and inflammation. miRNAs exert their role in the pathophysiology of atherosclerosis via the regulation of atherosclerosis-prone genes, as well as their impact in regulating post-transcriptional gene expression. By affecting the level of synthesized proteins within cells, they may be significant in driving the dysregulation that affects endothelial cells, VSMCs and leukocytes, which initiates and augments the growth of an atherosclerotic plaque [38].

The third step was to evaluate the interconnection of VSMCs with other cellular components of atheroma formation. Within the vessel wall, the communication between VSMCs and endothelial cells is essential not only for the development but also to the homeostasis of the mature blood vessel. Vascular injury arising from atherosclerosis is manifested by disruptions in endothelial cell-smooth muscle cell signalling [39,40]. Monocytes trapped in the subendothelial space may bind to VSMCs and undergo survival and differentiation through key signalling pathways and soluble and insoluble factors, e.g., those released by VSMCs [41].

Endothelial permeability, inflammation and atherosclerosis are inextricably linked. Dysfunction of the endothelial lining of lesion-prone areas of the arterial vasculature is an important contributor to the pathobiology of atherosclerotic cardiovascular disease, namely, endothelial cells are sentinels of cardiovascular health [39]. Once activated, endothelial cells set into motion a complex pathogenic sequence initially involving the selective recruitment of circulating monocytes from the blood into the intima, where they differentiate into macrophages and internalize modified lipoproteins to become foam cells [42]. EA.hy926 exposed to VSMCs^PCSK9^-EVs had a raised expression of adhesion molecules (e.g., vascular adhesion molecule-1, inter-cellular cell adhesion molecule-1, E-selectin), all features that drive vascular inflammation and atherosclerosis initiation and progression [43]. In said context, PCSK9 deficiency reduces endothelial dysfunction by reducing the expression of lower endothelial expression of the genes encoding, e.g., *ICAM-1*, *CCL2*, *IL-6* and *IL-1β* [44].

In line with this knowledge, VSMC^PCSK9^-EVs enhanced the migratory capacity of monocytes, a type of inflammatory cell that plays a role in atherosclerosis because of its ability to migrate into the arterial wall. PCSK9 directly facilitates inflammation in atherosclerotic lesions via the recruitment of monocytes, whereas PCSK9 inhibition ameliorates inflammation by reducing the CCR-2-related migratory ability of monocytes [45]. We found an increased expression of inflammatory cytokines in THP-1 treated with VSMCs^PCSK9^-EVs, as it was for the enrichment in pathways involved in immune response, immune effector process, response to stress and cellular response to cytokines. On this matter, PCSK9 seems intertwined with inflammation with implications in the atherosclerotic process [37,46,47]. A further support in favour of our hypothesis is given by the raised levels of the phosphorylated form of STAT3 in THP-1 exposed to VSMC^PCSK9^-EVs. STAT3 phosphorylation is markedly increased in atherosclerotic lesions; indeed, silencing the STAT3 pathway prevents atherosclerotic lesion formation, at least in murine models [48]. Consistently, the expression of SOCS3, a protein which plays negative feedback on STAT3, was abrogated [49]. The relationship among PCSK9, STAT3 and SOCS3 has been previously demonstrated by our group [50,51].

Other players in the development of atherosclerotic cardiovascular disease are macrophages. Predominantly derived from circulating monocytes and local proliferation, macrophage numbers increase exponentially within the aorta during atherogenesis [52]. Macrophages contribute to the maintenance of the local inflammatory response by secreting proinflammatory cytokines and chemokines. VSMC^PCSK9^-EVs favoured a pro-inflammatory milieu in macrophages. The role of PCSK9 in increasing human inflammatory signalling is supported by the observation that the release of the proinflammatory cytokines TNF-α and IL-1β is decreased in PCSK9-silenced macrophages [53], whereas PCSK9 treatment induces a pro-inflammatory response in macrophages [54]. In line with these findings, the overexpression of gain-of-function mutant of PCSK9 (D377Y) in *Ldlr^−/−^* mice induced pro-inflammatory macrophages activation (e.g., IL-1β, TNF-α and MCP-1 were raised) [55]. Furthermore, the internalization of PCSK9 in macrophages increased the production of reactive oxygen species and that of pro-inflammatory cytokines through a TLR4-dependent mechanism [56].

In this complex scenario, VSMC^PCSK9^-EVs reduced the migratory capacity of macrophages. Macrophages that accumulate in atherosclerotic plaques have diminished migratory capacity, hampering their capacity to resolve inflammation, and thus contribute to the formation of advanced, complex plaques [57]. The first robust evidence to support macrophage emigration from plaques was presented in pioneering pig studies from the 1970s to 1980s; interestingly, the rate of macrophage egress was found to decrease with atherosclerosis progression [58]. The reduced capacity to exit the plaque is a feature that comes hand-in-hand with the increased uptake of oxLDL. In the arterial intima, macrophages (highly phagocytic cells) internalize accumulated modified lipoproteins (e.g., oxLDL) through scavenger receptors [59]. We found that the uptake of oxLDL was raised in macrophages exposed to VSMC^PCSK9^-EVs with a concomitant upregulation of CD36, a multifunctional receptor whose expression is significantly increased in macrophages in human carotid atherosclerotic tissue, particularly in advanced stages of atherosclerosis [60]. On this matter, the silencing of PCSK9 suppresses the ox-LDL-induced up-regulation of proinflammatory cytokine (IL-1α, IL-6, and TNF-α) in THP-1-derived macrophages [61], whereas Inclisiran, a PCSK9 inhibitor, downregulates the expression of CD36 in macrophages [62].

Our in vitro data on macrophages were corroborated in zebrafish embryos, a useful in vivo experimental model for the study of inflammation. The innate immune system of zebrafish shares several features with the human one, recapitulating molecular pathways and immune cell-cell type present in humans [63]. The recruitment of macrophages in the site of injection of VSMC^PCSK9^-EVs compared to VSMC^WT^-EVs may be the confirmation that VSMC^PCSK9^-EVs carry an inflammatory chemoattractant feature, a hypothesis confirmed by the rise in the gene expression of *IL-1β* and *IL-8*. IL-1 has manifold effects on the cardiovascular system inducing inflammatory functions in human endothelial cells, stimulating adhesion molecules that recruit leukocytes, and promoting the expression of MCP-1, a chemoattractant for mononuclear phagocytes implicated in inflammatory cardiovascular diseases [64].

A further aspect that we took into account was the mitochondrial biogenesis. Mitochondrial dysfunction appears to be one of the key links in the pathogenesis of atherosclerosis and a target for therapeutic agents. Given the plethora of functions, it has been recognized that mitochondria are integrated into the cell signalling circuitry, and a modulation of these cellular pathways is at the basis of many inflammatory diseases such as atherosclerosis [65,66]. Specifically, VSMC^PCSK9^-EVs led to a drop in the oxidative phosphorylation and a rise in glycolytic activity in THP-1 monocytes. Cells involved in the pro-inflammatory response must rapidly provide energy to fuel inflammation, which is essentially accomplished by glycolysis and high lactate production [67,68].

## 4. Materials and Methods

### 4.1. Cell Cultures

Human vascular smooth muscle cells (VSMCs; A617 from human femoral artery) were maintained in Dulbecco’s Modified Eagle’s Medium (DMEM, Thermo Fisher Scientific, Milan, Italy) High Glucose supplemented by 10% Foetal Bovine Serum (FBS), penicillin/streptomycin solution (10,000 U/mL and 10 mg/mL, respectively), l-glutamine and sodium pyruvate (Merck, Milan, Italy) and maintained in humidified incubators at 37 °C and 5% CO_2_. VSMCs overexpressing human PCSK9 (VSMCs^PCSK9^) were generated by means of HEK293-Phoenix (φNX-A) cells as previously described [69], by using empty pBMN-IRES-puromycin vector or the same vector carrying human PCSK9 coding sequence to generate, respectively, VSMCs^PURO^ (VSMCs^WT^) and VSMCs^PCSK9^. For cell morphology evaluation, VSMCs were seeded in a 35 × 10 mm polystyrene non-pyrogenic Petri dish at a cell density of 0.4M cells/dish in DMEM supplemented with 10% FBS. Cell culture medium was replaced twice, at day 3 and day 5, and the morphology was evaluated on day 7.

THP-1 monocyte cells were cultured in Gibco Roswell Park Memorial Institute 1640 (RPMI, Thermo Fisher Scientific, Milan, Italy) supplemented with 10% FBS and penicillin (10,000 U/mL), streptomycin (10 mg/mL) and β-mercaptoethanol 0.05 mM (Merk, Milan, Italy). THP-1-derived macrophages were obtained by treating THP-1 monocytes with phorbol 12-myristate 13-acetate (PMA; 10 ng/mL, Merk, Milan, Italy) for 72 h.

The endothelial cell line EA.hy926 was cultured in DMEM supplemented with penicillin (10,000 U/mL), streptomycin (10 mg/mL), non-essential amino acids, tricine buffer (Merk, Milan, Italy), HAT (Merk, Milan, Italy), HEPES (Euroclone, Milan, Italy) and 10% FBS on gelatin-coated 6-well plates.

J774 macrophages were cultured in DMEM High glucose supplemented with 10% FBS and penicillin (10,000 U/mL), streptomycin (10 mg/mL), L-glutamine and maintained in humidified incubators at 37 °C and 5% CO_2_.

Relative to treatments with isolated EVs, all cell lines were starved overnight in RPMI or DMEM supplemented with 0.1% bovine serum albumin (BSA). After starvation, media were replaced, and cells were treated with 5 × 10^8^ EVs/mL for 24 h. According to the MISEV guidelines, since even a small percentage of cell death could release cell membranes that outnumber true released EVs, the percentage of apoptotic cells was tested [70]. Cell viability was assessed by using Dead Cell Apoptosis Kits with Annexin V (Invitrogen, ThermoFisher Scientific) in cell lines supplemented with FBS or BSA (described in detail at Section 4.8).

### 4.2. Extracellular Vesicles Isolation

According to the most recent MISEV guidelines [70], to isolate EVs, VSMCs^WT^ and VSMCs^PCSK9^ were grown in flasks 75 cm^2^ (Euroclone, Milan, Italy) until 80% confluence. Then, the cell medium was replaced with 12 mL of serum-free DMEM. After 24 h, the conditioned medium was collected, cells harvested and counted. The medium was centrifuged at 1000-, 2000- and 3000× *g* for 15 min at 4 °C, and the pellet from each centrifugation was discarded to remove cells and debris. Supernatant was transferred to 10.4 mL polypropylene ultracentrifuge tubes (Beckman Coulter, CA, USA), and ultracentrifuged at 110,000× *g* for 4 h, at 4 °C. To minimize the background contribution of interfering particles, two pellets from the same cell line were resuspended in 500 μL of Dulbecco’s Phosphate Buffered Saline without calcium chloride and magnesium chloride (PBS, Merck, Milan, Italy) triple filtered through a 0.10 μm pore-size polyethersulfone filter. All ultracentrifugation steps were performed on an Optima MAX-XP ultracentrifuge (Beckman Coulter, CA, USA) with a type 55-MLA Rotor (Beckman Coulter).

### 4.3. Evaluation of the Uptake of EVs

EVs were labelled by using lipophilic dye PKH67 Fluorescent Cell Linker Kit (Merk, Milan, Italy). One microliter of PKH67 ethanolic dye solution was added to 250 μL of Diluent C and incubated 5 min with a pellet of EVs. The labelling reaction was stopped by adding an equal volume of 1% BSA for 1 min to allow the binding of excess dye. EVs were ultracentrifuged at 110,000× *g* for 2 h to remove the free dye. The supernatant was discarded, and the EV pellet was resuspended in 500 μL of PBS triple-filtered through a 0.10 μm pore-size polyethersulfone filter. Recipient cells (e.g., THP-1 monocytes) were seeded at a cell density of 2 × 10^6^/well in a 6-well plate and exposed to labelled EVs for 24 h. Cells were then collected and centrifuged to remove unincorporated EVs and fluorescence (λex 490 nm; λem 502 nm) was analysed by high-resolution flow cytometry (NovoCyte Flow Cytometer, Agilent, Santa Clara, CA, USA).

### 4.4. Nanosight Tracking Analysis (NTA)

The number and dimension of EVs were assessed by NTA by using the NanoSight NS300 system (Malvern Panalytical, Malvern, UK), which measures the Brownian motion of particles suspended in fluid and displays them in real-time through a high sensitivity CCD camera. For NTA analysis, PBS-resuspended EVs were properly diluted in order to reach a number of particles per frame between 20–120. Five videos of 30 s were recorded for each sample and data were analysed with NTA software (Malvern Panalytical), providing high-resolution particle-size distribution profiles as well as measurements of the EV concentrations.

### 4.5. Transmission Electron Microscopy (TEM)

After isolation, EVs resuspended in PBS and properly diluted were incubated for 5 min onto carbon-coated copper grids, 200 mesh (Electron Microscopy Sciences, Hatfield, PA, USA) at room temperature. Once adsorbed on the grids, EVs were fixed with 2% glutaraldehyde in PBS (Electron Microscopy Sciences, Hatfield, PA, USA) for 10 min, then washed 3 times in Milli-Q water; negative staining was performed with 2% phosphotungstic acid; finally, the grids were air dried and observed using a Microscope Zeiss GEMINI 500.

### 4.6. qPCR

Total mRNA was extracted by using spin column (Qiagen, Milan, Italy) according to the manufacturer’s instructions. Reverse transcription-polymerase first-strand cDNA synthesis was performed by using the Maxima First Strand cDNA synthesis kit (Thermo Fisher Scientific). Quantitative PCR was then performed by using the Thermo SYBR Green/ROX qPCR Master Mix kit (Thermo Fisher Scientific) and specific primers for selected genes. The analyses were performed with the 9600 Bio-Rad Real-Time PCR Detection Systems (Bio-Rad Laboratories, Milan, Italy). PCR cycling conditions were as follows: 95 °C for 10 min, 40 cycles at 95 °C for 15 s, and 55 °C for 1 min. Data were expressed as Ct values and used for the relative quantification of targets with the 2^−ΔΔCt^ calculation. In the case of endothelial cell line cells, THP-1 and THP-1-derived macrophages, 18S and 36B4 were used, respectively, as housekeeping genes. Primer sequences used for qRT-PCR analysis are shown in Appendix A.

Relative to zebrafish, total RNA was extracted from at least 20 systemically EVs-treated embryos at 20 h post-injection (hpi) using NucleoZOL reagent (Macherey-Nagel, Düren, Germany) according to the producer’s instructions. The concentration and purity of RNA were measured using NTA. To avoid possible genomic contamination, RNA was treated with RQ1 RNase-free DNase (Promega, Madison, WI, USA). 1 μg of DNase-treated RNA was reverse-transcribed with GoScript Reverse Transcription Kit (Promega, Madison, WI, USA), using a mixture of random primers and oligo(dT), following the manufacturer’s instructions. qPCR analyses were performed with the GoTaq qPCR Master Mix (Promega), in the QuantStudio 5 Real-Time PCR System (Applied Biosystems, ThermoFisher Scientific), following the manufacturer’s guidelines. The calculation of gene expression was based on the 2^–∆∆Ct^ method [71]. *rpl8* and β-*actin* were used as the internal reference genes, for normalization purposes. Primers sequences used for qRT-PCR analysis are shown in Appendix A.

### 4.7. Western Blot

Total cytosolic protein extracts of EVs were obtained by resuspending EV pellets in 50 μL of RIPA buffer (0.05M Tris-HCl pH 7.7, 0.15 M NaCl, 0.8% TritonX-100, 0.8% sodium deoxycholate, and 0.08% SDS, 10 mM EDTA, 100 μM sodium vanadate, 50 mM NaF, 5 mM Iodoacetic acid) containing a cocktail of protease and phosphatase inhibitors (Roche Diagnostics, Rotkreuz, Switzerland). After 1 h on ice, EV lysates were centrifuged at 14,000× *g* for 10 min. Total cytosolic protein extracts of THP-1 and THP-1-derived macrophages were obtained by collecting cells in 70 μL of Mammalian Protein Extraction Reagents (Thermo Fisher Scientific) containing a cocktail of protease and phosphatase inhibitors (Roche Diagnostics). Protein concentration was determined by the Pierce BCA protein assay (Thermo Fisher Scientific). Twenty micrograms of proteins and a molecular mass marker (Novex Sharp Protein Standard, Invitrogen; Thermo Fisher Scientific) were separated on a 10% SDS-PAGE gel under denaturing and reducing conditions. Proteins were then transferred to a nitrocellulose membrane at 200 mA for 120 min. The membranes were washed with Tris-buffered saline-Tween 20, and nonspecific binding sites were blocked in Tris-buffered saline-Tween 20 containing 5% BSA (Sigma-Aldrich) for 90 min at room temperature. The blots were incubated overnight at 4 °C with a diluted solution (5% BSA or nonfat dry milk) of the human primary antibodies that are shown in Appendix A. Membranes were washed with Tris-buffered saline-Tween 20 and then exposed for 90 min at room temperature to a diluted solution (5% nonfat dry milk) of the secondary antibodies (anti-mouse and anti-rabbit peroxidase-conjugated secondary antibodies; New England Biolabs, MA). Immunoreactive bands were detected by exposing the membranes to Clarity Western ECL chemiluminescent substrates (Bio-Rad Laboratories) for 5 min, and images were acquired with a ChemiDoc XRS System (Bio-Rad Laboratories). Densitometric readings were evaluated using the ImageLab software version 6.0.1 (Bio-Rad Laboratories).

### 4.8. Enzyme-Linked Immunosorbent Assay

PCSK9 concentrations were measured in cell culture media of VSMCs^WT^ and VSMCs^PCSK9^ by a commercial ELISA kit (R&D Systems, Minneapolis, MN, USA). Samples were incubated onto a microplate pre-coated with a monoclonal human-PCSK9-specific antibody. Sample concentrations were obtained by a four-parameter logistic curve-fit, with a minimum detectable PCSK9 concentration of 0.219 ng/mL and were normalized by protein concentrations.

### 4.9. Flow Cytometry

EVs were characterized by high-resolution flow cytometry (MACSQuant, Miltenyi Biotec, Bergisch Gladbach, Germany), according to a previous protocol [12]. Briefly, sample acquisition was performed at the minimum speed flow (25 µL/min) using a MACSQuant Analyzer (Miltenyi Biotec). Sheath fluid was filtered through a 0.1 μm pore size filter to further improve the signal-to-noise ratio. The fluorescent beads Fluoresbrite^®^ YG Carboxylate Microspheres Size Range Kit I (0.1, 0.2, 0.5, 0.75, and 1 μm; Polysciences Inc, Warrington, PA, USA) were used to set the calibration gate in the FSC/FL1 and FSC/SSC dot plots. Using a side scatter (SSC) threshold of 10 arbitrary units, the lower sensitivity of the instrument was determined and the SSC and FITC voltages were set up. An overlap in the 100 nm beads population and the background noise were observed. In this way, it was possible to gate the EVs ≥ 200 nm diameter. A total of 30 μL of sample was acquired on the MACSQuant Analyzer. Event numbers, analysed at a low flow rate and below 10,000 events/second of equal sample volumes were counted. To assess the integrity of isolated EVs, 60 µL of sample were stained with 0.2 mM 5(6)-carboxyfluorescein diacetate N-succinimidyl ester (CFSE, Thermo Fisher Scientific) at 37 °C for 20 min. The immunophenotyping of EVs was analysed by means of tetraspanins family antibodies, purchased from Miltenyi Biotec and used according to the product manufacturers: CD9-APC (clone REA1071), CD63-APC (clone REA1055). The CFSE stained samples were incubated with 6 µL of CD9 and CD63 antibodies previously diluted 1:5 for 30 min in the dark at 4 °C. Each antibody aliquot was previously centrifuged at 17,000× *g* for 30 min at 4 °C to eliminate aggregates. To detect the autofluorescence of each antibody, 60 µL of triple 0.10 µm pore size membrane-filtered PBS (control sample) were stained with CFSE, CD9 and CD63. Quantitative multiparameter analysis of flow cytometry data was carried out using FlowJo Software (Ashland, OR, USA).

Relative to the apoptosis, cells were washed with PBS, centrifuged and stained with FITC annexin V and red fluorescent propidium iodide (Thermo Fisher). All the samples were analysed by flow cytometry (Novocyte 3000, ACEA Bioscience, Santa Clara, CA, USA), measuring the fluorescence emission at 530 nm and >575 nm.

### 4.10. Migration Assay

Cell migration of THP-1 monocytes and J774 macrophages was assessed by using a 24-transwell support of 5 μm pore size (Costar, Cambridge, MA, USA). THP-1 and J774 cells were seeded in 6-well plates at a density of 1.2 × 10^6^ cells/well in RPMI + 0.1% BSA (THP-1) and 3 × 10^5^ in DMEM + 0.1% BSA (J774) and were treated with 5 × 10^8^ EVs/mL isolated from both VSMCs^WT^ and VSMCs^PCSK9^ for 24 h. The lower chamber was filled with 600 µL of RPMI containing CCL2 at a concentration of 10 ng/mL. Plates were incubated at 37 °C with 5% CO_2_ for 10 h (THP-1) and 24 h (J774). At the end of the assay, cells migrated to the lower chamber were counted using an optical microscope.

### 4.11. Preparation of Oxidized LDL (oxLDL) and Lipoprotein-Induced Cholesterol Accumulation

Chemical oxidation of human LDL (Sigma Aldrich) was performed under sterile conditions as previously described [72], by incubating LDL at 37 °C for 24 h, at 0.2 mg protein/mL phosphate buffer saline (PBS) + 20 µM CuSO_4_. Oxidation was blocked in ice, with the addition of 40 μM butylhydroxytoluene (BHT). Modification of lipoproteins was verified by the change in the electrophoretic mobility tested by non-denaturing gel electrophoresis as compared to native LDL [73].

Human THP-1 monocytes were grown in RPMI containing 10% FBS (Euroclone, Milano, Italy) in the presence of antibiotics (penicillin–streptomycin, Thermo Fisher Scientific, USA). Cells were plated in the presence of 10 ng/mL phorbol 12-myristate 13-acetate (Sigma-Aldrich, Milano, Italy) for 72 h to allow differentiation into macrophages. After overnight starvation in RPMI supplemented with 0.1% BSA, cells were incubated for 24 h with 5 × 10^8^ EVs/mL isolated from VSMCs^PCSK9^ and from VSMCs^WT^. Cells were then incubated for additional 24 h with RPMI + 1% FBS in the absence or presence of 50 µg/mL of native LDL or oxLDL. At the end of the incubation, cell monolayers were lysed in 1% sodium cholate solution (Sigma-Aldrich) and supplemented with 10 U/mL DNase (Sigma-Aldrich). Cholesterol was then measured fluorometrically using the Amplex Red Cholesterol Assay Kit (Molecular Probes, Eugene, OR, USA) following manufacturer’s instructions. An aliquot of cell lysates was used to measure cell proteins by the BCA assay (Thermo Fisher Scientific). Intracellular cholesterol content was expressed as micrograms of cholesterol/milligram of protein.

### 4.12. Mitochondrial Bioenergetic Evaluation

Mitochondrial activity was measured by the Agilent Seahorse XF Cell Mito Stress Test which measures key parameters of mitochondrial function by directly measuring the oxygen consumption rate (OCR) of cells. OCR was measured at baseline to determine basal respiration and after injection of optimised doses of specific mitochondrial activators and inhibitors to determine non-mitochondrial respiration, maximal respiration and spare respiratory capacity. Briefly, after 24 h-treatment with EVs, THP-1 cells were plated on a pre-coated Seahorse XF24-well microplate at a density of 4 × 10^5^ cells/well in 100 µL of Seahorse XF DMEM, supplemented with glucose, pyruvate and glutamine. The plate was centrifuged at 200× *g* for 5 min and filled to 500 µL with Seahorse XF DMEM. The plate was incubated at 37 °C in a non-CO_2_ incubator for 45 min. The OCR was measured using the XFe24 analyser, with 3 baseline measurements recorded before and after adding Oligomycin (ATP synthase inhibitor; 1 μM), carbonyl cyanide p-trifluoromethoxy-phenylhydrazone (FCCP, an uncoupling agent, 1 μM), and a mix of Rotenone and Antimycin A (complex I and III inhibitors, respectively; 0.5 μM) [74]. To further shed light on THP-1 glycolysis, the Seahorse XF Glycolytic Rate Assay was used. Rates at basal level as well as after injections of Rotenone plus antimycin A (1 μM) and 2-deoxy-D-glucose (50 mM) were recorded [68].

### 4.13. Mass Spectrometry Analysis

For proteomic analysis, samples were dissolved in 25 mmol/L NH_4_HCO_3_ containing 0.1% RapiGest (Waters Corporation, Mildford, CT, USA), sonicated and centrifuged at 13,000× *g* for 10 min. Fifty μg of proteins were incubated 15 min at 80 °C and reduced with 5 mmol/L DTT at 60 °C for 15 min, followed by carbamidomethylation with 10 mmol/L iodoacetamide for 30 min at room temperature in the darkness. Then, 2.5 μg of sequencing grade trypsin (Promega) was added to each sample and incubated overnight at 37 °C. After digestion, 2% TFA was added to hydrolyze RapiGest and to inactivate trypsin. Proteomic analysis of secretome samples was performed as described above after desalting, concentration and digestion as previously described [75]. Label-free mass spectrometry analysis, LC-MSE, was performed on a hybrid quadrupole-time of flight mass spectrometer (Synapt XS, Waters corporation, Milford, CT, USA) coupled to a UPLC Mclass system and equipped with a nanosource (Waters Corporation, Milford, CT, USA). Samples were injected into a Symmetry C18 nanoACQUITY trap column, 100 Å, 5 μm, 180 μm × 2 cm (Waters Corporation, Milford, MA, USA) and subsequently directed to the analytical column HSS T3 C18, 100 Å, 1.7 μm, 75 μm × 150 mm (Waters Corporation, Milford, MA, USA), for elution at a flow rate of 300 nL/min by increasing the organic solvent B concentration from 3 to 40% over 90 min, using 0.1% *v*/*v* formic acid in water as reversed phase solvent A, and 0.1% *v*/*v* formic acid in acetonitrile as reversed phase solvent B. All the analyses were made in triplicate and analysed by LC-MSE [76], with the introduction of an ion mobility-enhanced data-independent acquisition (IMS-DIA). The spectral acquisition time in each mode was 0.5 s, with a 0.1 s inter-scan delay. In the low-energy MS mode, the data were collected at constant collision energy of 6 eV; in high energy mode, fragmentation was obtained by applying drift time-specific collision energies [77]. The software Progenesis QI for proteomics (Version 4.0, http://www.nonlinear.com, accessed on 1 September 2022) was used for the quantitative analysis of peptide features and protein identification [78].

### 4.14. Gene Ontology Analysis

Proteomics data were analysed with the Search Tool for the Retrieval of Interacting Genes/Proteins (STRING 10.5) database to identify enriched gene ontology terms in the biological process, molecular function or cellular component categories. In particular, the enrichment function of STRING that calculates an enrichment *p*-value based on the Hypergeometric test using the method of Benjamini and Hochberg for correction of multiple testing (*p* value cut-off of <0.05) was used.

### 4.15. EV-miRNAs Isolation and Analysis

To prepare the EV pellets for miRNA extraction, supernatant was ultracentrifuged (Beckman Coulter Optima-MAX-XP) at 110,000× *g* for 4 h, at 4 °C and decanted. The EV pellet was stored at −80 °C until use. Isolation of miRNAs from EVs was performed with the combination of miRNeasy kit and RNeasy Cleanup Kit (Qiagen), according to the manufacturer’s protocol. miRNAs were eluted in 20 µL of Nuclease-Free Water and stored at −80 °C, until use. miRNAs reverse transcription (RT) and preamplification reactions, followed by real-time RT-PCR analysis with the QuantStudio™ 12K Flex OpenArray^®^ Platform (Applied Biosystems, Milan, Italy), were previously described. Gene Expression Suite Software (Applied Biosystems) was used to process miRNA expression data from the “TaqMan™ OpenArray™ Human MicroRNA panel” (ThermoFisher) analysis.

To elucidate the possible role of the EV-miRNAs, we performed a miRNA target analysis using SpidermiR by R software (v 4.0.4). Then, we compared genes obtained by the miRNA target analysis, with genes associated with atherosclerosis and inflammation downloaded from DisGeNET v 7.0. In addition, for each miRNA, we performed an enrichment analysis [79].

### 4.16. Zebrafish Husbandry

Zebrafish (*Danio rerio*) were maintained at the Università degli Studi di Milano according to international (EU Directive 2010/63/EU) and national guidelines (Italian decree No 26 of the 4th of March 2014). Embryonic ages are expressed in hours post fertilization (hpf) and days post fertilization (dpf). Embryos were collected by natural spawning, staged according to Kimmel et al. [80] and raised at 28.5 °C in fish water (Instant Ocean, 0,1% Methylene Blue) in Petri dishes, according to established techniques. After 24 hpf, to prevent pigmentation, 0.003% 1-phenyl-2-thiourea (PTU, Sigma-Aldrich, Saint Louis, MO) was added to the fish water. Embryos were washed, dechorionated and anaesthetized, with 0.016% tricaine (Ethyl 3-aminobenzoate methanesulfonate salt, Sigma-Aldrich) before observations and microinjection.

### 4.17. Microinjection of Zebrafish Embryos and Immunostaining Analysis

Two or 3 dpf embryos were microinjected systemically or locally with VSMC^PCSK9^- or VSMC^WT^-derived EV suspensions. Before being injected, aliquots of EV suspensions were stained with 0.2 μM CFSE. For immune response experiments, 2 nL of a suspension of EVs was systematically microinjected into embryos at 2 dpf [81]. At least 20 embryos were injected for each treatment, and each experiment was repeated at least twice. For macrophage recruitment experiments, 72 hpf embryos were injected with 2 nL of a suspension of EVs in the hindbrain ventricle and intramuscularly, as previously described [26]. For intra-muscle injection, EV suspension was delivered among the second and the fifth somite before yolk extension. Embryos were kept at 28.5 °C until their use for analysis.

Evaluation of macrophage recruitment at the injection site was performed by immunofluorescence staining of zebrafish embryos as described in [82] with some modifications. Embryos were fixed at the indicated developmental stage for 2 h in 4% paraformaldehyde (PFA; Merck) in Phosphate Buffer Saline (PBS; Merck) overnight at 4 °C, then rinsed in PBS and washed in PBT (PBS, 1% Tween-20, Sigma Aldrich, Milan, Italy). Embryo permeabilization was performed in cold acetone for 25 min at room temperature. Embryos were then blocked for 3 h, at RT in a solution of PBT plus 5% BSA. Subsequently, embryos were incubated with the primary antibody in blocking solution overnight at 4 °C with agitation. Successively, embryos were washed in PBT over 2 h, at RT and then incubated in blocking solution for 2 h at room temperature. Embryos were then incubated with the secondary antibody in blocking solution, at 4 °C with agitation. Fluorescent images were sequentially acquired using an epi-fluorescence stereomicroscope (M205FA, Leica, Wetzlar, Germany) mounting TRITC and GFP filters (excitation of 542 and 469 nm). When necessary, images were processed using the Adobe software. Macrophage count/quantification was measured in an indicated dashed line box, by means of ImageJ software (Developer: W. Rasband).

### 4.18. Statistical Analysis

Data are given as means ± SEM of three independent experiments. When possible, *p* values were determined by t-test or Mann–Whitney. Otherwise, differences between treatment groups were evaluated by one-way analysis of variance. A probability value of *p* < 0.05 was considered statistically significant. Statistical analysis was performed using the Prism statistical analysis package version 8.0 (GraphPad Software, San Diego, CA, USA). Concerning EVs, descriptive statistics were performed on all variables. Linear regression analyses were applied to evaluate the association between mean EV size produced by VSMCs^WT^ and VSMCs^PCSK9^. EV size showed a skewed distribution and was naturally log-transformed to achieve normal distribution. We reported geometric means with 95% CI. Negative binomial linear regression models were applied to evaluate the association between EV total average number produced by VSMCs^WT^ and VSMCs^PCSK9^. We tested over-dispersion by the likelihood ratio test, and based on its results, we decided to apply the negative binomial models instead of Poisson regression. We reported marginal means with 95% CI and *p*-values. For each EV size, we estimated EV mean concentration and 95% CI for EVs^PCSK9^, with negative binomial linear regression models. Due to the high number of comparisons, we used a multiple comparison method based on Benjamini–Hochberg False Discovery Rate (FDR) to calculate the FDR *p*-value. To display results of the analyses we used a series graph for EV mean concentrations of each group and vertical bar charts to represent FDR *p*-values and *p*-values. For the two graphs, the X axis was the size of EVs. Statistical analyses were performed with SAS 9.4 software (SAS Institute Inc., Cary, NC, USA).

## 5. Conclusions

At the conclusion of this work, our data must be interpreted in a frame of limitations. First, although cholesterol has an impact on the fate of EVs and on their uptake by target cells, we did not evaluate the cholesterol content of EVs [83]. However, we did not detect any change in EV concentrations, size and shape. Second, we did not specifically evaluate the biodistribution of EVs in embryos of zebrafish. To overcome this drawback, we showed that sulfo-cyanine 7.5-stained EVs, systematically injected in the embryos of zebrafish, were concentrated in the caudal region where they encountered components of innate immunity (Appendix A). Third, instead of using murine J774 macrophages, the human U937 could have been used. However, these two cell lines are readily exchangeable when evaluating phagocytosis, cytokine production and morphological analyses [84]. Fourth, to disentangle the role of CD36 in the uptake of oxLDL, an experiment in which CD36 was silenced could have been performed.

In conclusion, the results of the present study describe a new indirect role played by PCSK9 in atheroma formation by providing new evidence to the hypothesis that PCSK9 is involved in the pathogenesis of cardiovascular disease, independent of hyperlipidaemia but possibly by fuelling a pro-inflammatory feed-forward-loop. By influencing the cargo of EVs released from VSMCs, PCSK9 favours a pro-inflammatory milieu among cell components of atheroma formation. However, to further disentangle the liaison between PCSK9 and EVs in the context of atheroma formation, future studies (also epidemiological) should evaluate the impact of PCSK9 inhibitors on the EV release and cargo, specifically those of VSMC origin.

## Figures and Tables

**Figure 1 ijms-23-13065-f001:**
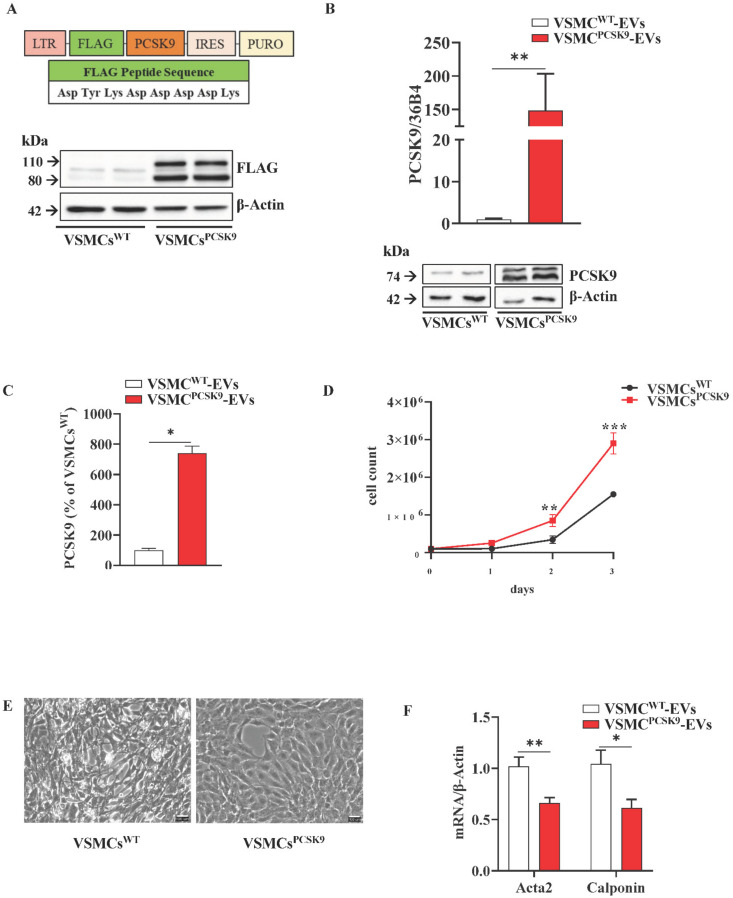
Characterization of vascular smooth muscle cells (VSMCs). (**A**) Generation of VSMCs overexpressing PCSK9 by using a pBMN-IRES-puromycin vector carrying human PCSK9 coding sequence. FLAG peptide has the following sequence Asp-Tyr-Lys-Asp-Asp-Asp-Asp-Lys; PURO stands for puromycin; IRES stands for internal ribosome entry site; LTR stands for long terminal repeat. Protein expression of FLAG was assessed by WB analysis. β-actin was used as a housekeeping protein. Representative image of three independent experiments. (**B**) Gene (quantitative PCR) and protein expression (WB) of PCSK9 in VSMCs. 36B4 was used as a reference gene and β-Actin was used as a housekeeping protein. For WB, the reported blot is representative of three independent experiments. (**C**) Data obtained from the ELISA assay are expressed as % of PCSK9 released in the cell culture medium of VSMCs^PCSK9^ compared to VSMCs^W**T**^; the results were normalized for the total protein content. n = 6 per group. (**D**) Proliferative assay of VSMCs (cell counting). n = 6 per group. (**E**) VSMCs were cultured in standard conditions for seven days and the morphology was evaluated by optical microscopy [17]. (**F**) Gene expression of VSMC markers(Acta2 and Calponin). β-Actin was used as a reference gene. n = 6 per group. (**G**) Gene ontology enrichment analysis relative to proteins with a significantly higher expression in VSMCs^PCSK9^. Red bar represents VSMCs^PCSK9^ and white bar represents VSMCs^W**T**^. Results are expressed relative to the normal control and as mean ± SEM. Differences between groups have been assessed by *t*-test. * *p* < 0.05, ** *p* < 0.01, *** *p* < 0.001 vs. control. Acta2, alpha-actin-2; PCSK9, proprotein convertase subtilisin/kexin type 9; VSMCs, vascular smooth muscle cell; VSMCs^PCSK9^, vascular smooth muscle cells overexpressing PCSK9; VSMCs^W**T**^, vascular smooth muscle cells wildtype; WB, Western blot.

**Figure 2 ijms-23-13065-f002:**
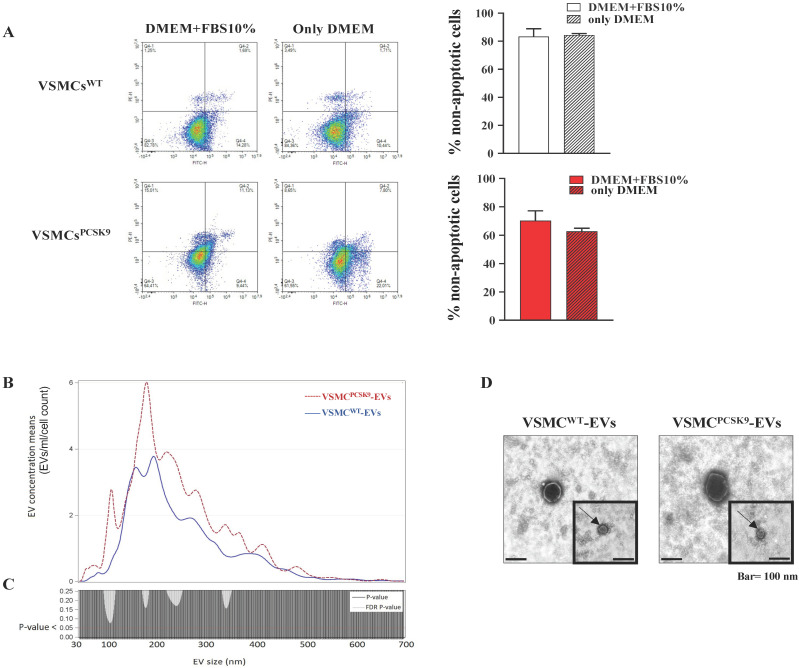
Phenotypic characterization of extracellular vesicles (EVs). (**A**) Cell viability of VSMCs supplemented or not with FBS was assessed by annexin V/propidium iodide double staining. n = 9 per group. (**B**) EV concentrations are expressed as means for each size for the VSMC^PCSK9^-EVs (dashed line) and VSMC^WT^-EVs (solid line). EV concentrations were calculated as marginal means from negative binomial regression models. n = 6 derived from 6 different experiments. (**C**) *p*-value and False Discovery Rate *p*-value of the comparisons of each size EV for the entire 30–700 nm size range are reported. n = 6 derived from 6 different experiments. (**D**) Representative ultrastructural images of EVs by TEM. For both, the main image shows an EV whose size is compatible with intermediate EVs, and the insert shows an EV whose size is compatible with small EVs (arrow). Scale bar = 100 nm. (**E**) The expression of tetraspanins CD9 and CD63 was evaluated by flow cytometry analysis in EVs. Representative panels of three independent experiments. (**F**) Protein expression of β1-integrin and Alix as assessed by WB in EVs. Representative blots of three independent experiments. (**G**) PCSK9 expression in VSMC^WT^-EVs and VSMC^PCSK9^-EVs through WB analysis. Representative blot of three independent experiments. VSMC^PCSK9^-EVs are those released by VSMC^PCSK9^ and VSMC^WT^-EVs are those released by VSMCs^WT^. EV, extracellular vesicles; FBS, foetal bovine serum; VSMC^PCSK9^-EVs, EVs released by VSMCs overexpressing PCSK9; VSMC^WT^-EVs, EVs released by VSMCs wildtype; TEM, Transmission electron microscopy; WB, Western blot.

**Figure 3 ijms-23-13065-f003:**
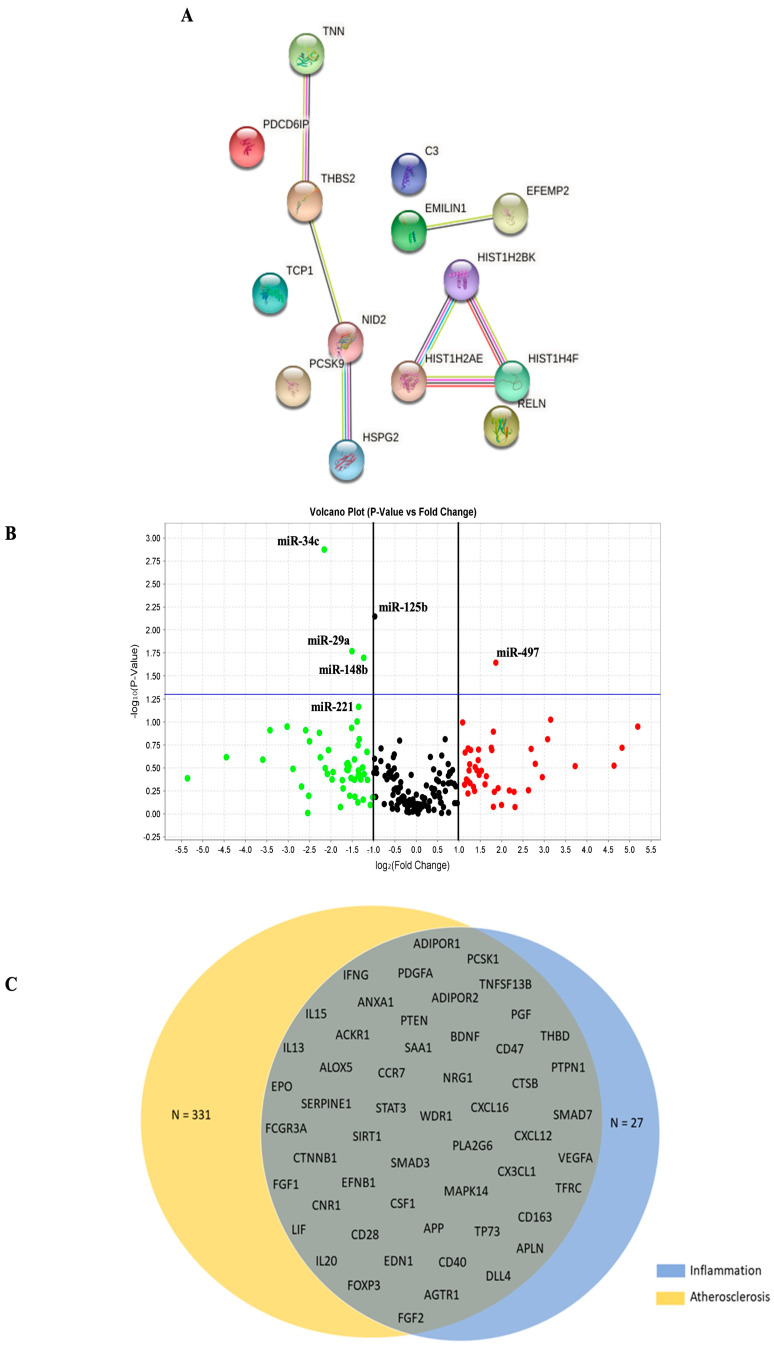
Characterization of EVs cargo. (**A**) Gene ontology enrichment analysis relative to proteins with a significantly higher expression in VSMC^PCSK9^-EVs. (**B**) Volcano plot of miRNA with a significantly different expression between VSMC^PCSK9^-EVs and VSMC^WT^-EVs. (**C**) Venn diagram of miRNAs targets. All predicted genes for the differentially expressed miRNAs were filtered according to their relationship with atherosclerosis and inflammation and were selected for checking gene targets overlap. Results show t miRNAs target common genes. EVs, extracellular vesicles; VSMC^PCSK9^-EVs, EVs released by VSMCs overexpressing PCSK9; VSMC^WT^-EVs, EVs released by VSMCs wildtype.

**Figure 4 ijms-23-13065-f004:**
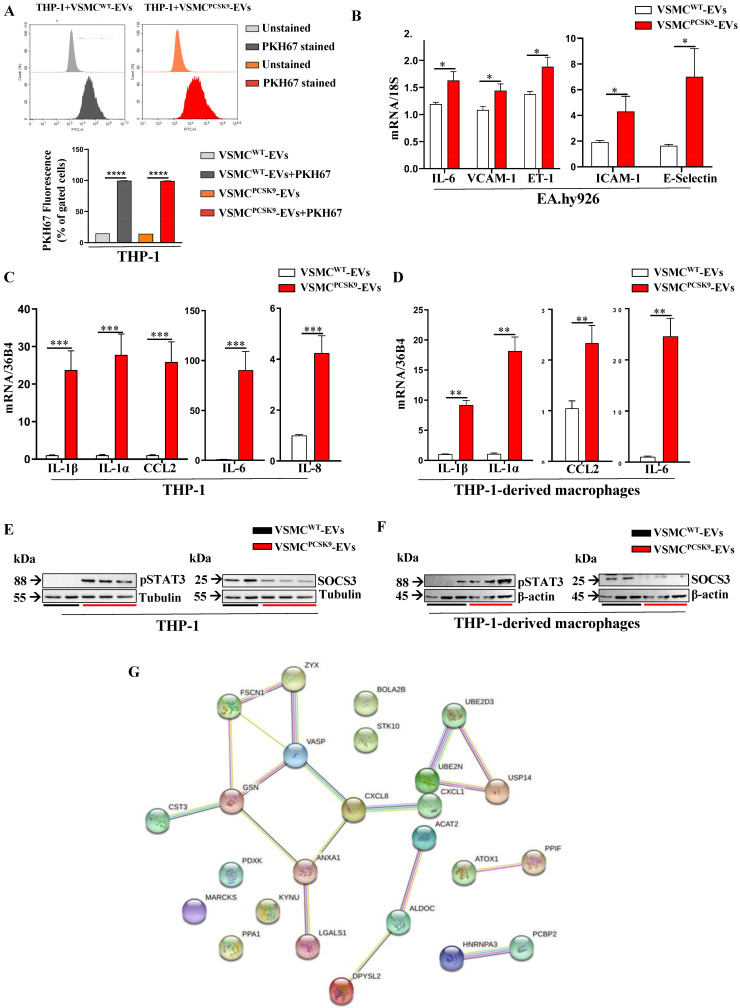
Impact of extracellular vesicles (EVs) on endothelial cells, THP-1, and THP-1-derived macrophages. (**A**) Plots and bar graphs show the uptake of VSMC^PCSK9^-EVs and VSMC^WT^-EVs by target cells (THP-1) assessed by flow cytometry. n = 3 per group. (**B**) Gene expression of pro-inflammatory cytokine IL-6 and cell adhesion molecules (VCAM-1, ET-1, ICAM-1 and E-selectin) in EA.hy926 endothelial cells. n = 6 per group. (**C**,**D**) Gene expression of pro-inflammatory cytokines and chemokines (IL-1β, IL-1α, IL-6, IL-8 and CCL2) in THP-1 monocytes (**C**) and in THP-1 derived macrophages (**D**) after treatment with VSMC^PCSK9^-EVs and VSMC^WT^-EVs. 18S (**B**) and 36B4 (**C**,**D**) were used as reference genes. n = 6 per group. E-F) Protein expression of pSTAT3 and SOCS3 as assessed by WB analysis in THP-1 monocytes (**E**) and THP-1 derived macrophages (**F**). Tubulin (**E**) and β-actin (**F**) were used as housekeeping proteins. Representative blot of three independent experiments. (**G**) Proteomic and GO analysis of secreted proteins from THP-1 cells incubated with VSMC^PCSK9^-EVs. Red line and bar represent cells exposed to VSMC^PCSK9^-EVs; black line and white bar represent cells exposed to VSMC^WT^-EVs. Results are expressed relative to the normal control and as mean ± SEM. Differences between groups have been assessed by *t*-test. * *p* < 0.05, ** *p* < 0.01, *** *p* < 0.001, *****p* < 0.0001 vs. control. EVs, extracellular vesicles; VSMC^PCSK9^-EVs, EVs released by VSMCs overexpressing PCSK9; VSMC^WT^-EVs, EVs released by VSMCs wildtype; SEM, standard error of the mean.

**Figure 5 ijms-23-13065-f005:**
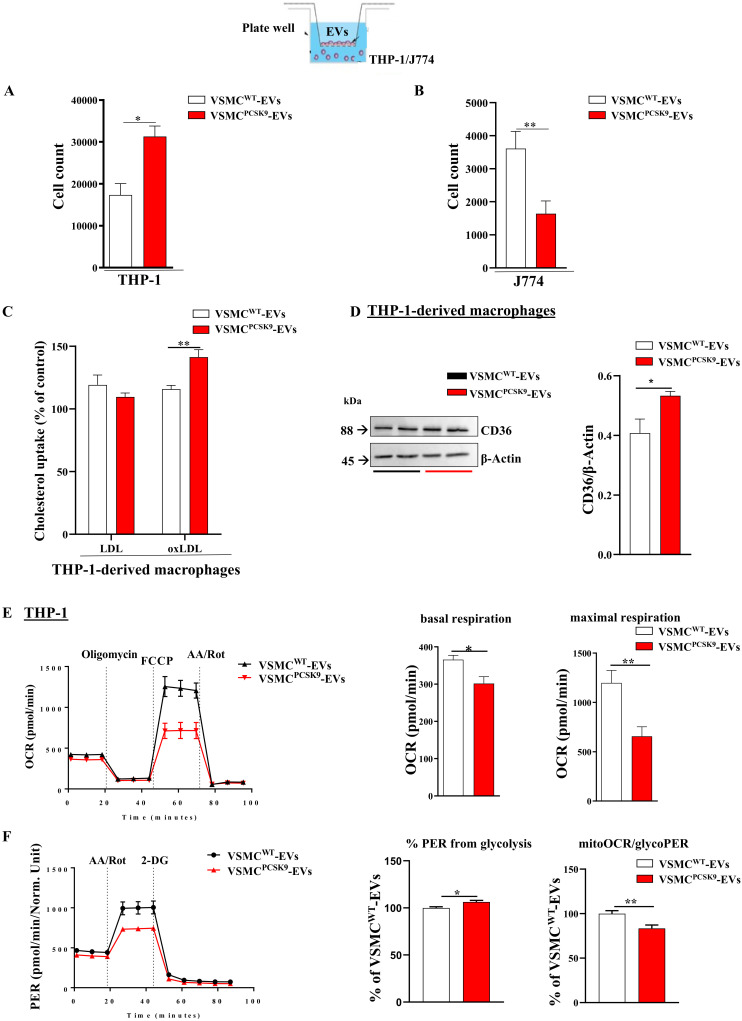
Cell migration, uptake of ox-LDL and mitochondrial respiration. (**A**) Migration of THP-1 upon exposure to EVs. n = 4 per group. (**B**) Migration of J774 macrophages upon exposure to EVs. n = 4 per group. (**C**) Uptake of oxLDL-derived cholesterol in THP-1 cells treated with EVs. n = 3 per group. (**D**) CD36 protein expression in THP-1-derived macrophages after exposure to VSMC^PCSK9^-EVs compared with VSMC^WT^-EVs as assessed by WB. β-actin was used as a housekeeping protein. Representative blot of three independent experiments. (**E**,**F**) Mitochondrial respiration function was assessed by Seahorse XFe24 analyser. Mitostress analysis (**E**) and glycolytic rate assay (**F**) were performed. Red line represents THP-1 exposed to VSMCs^PCSK9^-EVs; black line represents THP-1 exposed to VSMC^WT^-EVs. Results are expressed relative to the VSMC^WT^-EVs control. n = 5 per group. Differences between groups have been assessed by *t*-test. * *p* < 0.05, ** *p* < 0.01, versus control. EV, extracellular vesicles; VSMC^PCSK9^-EVs, EVs released by VSMCs overexpressing PCSK9; VSMC^WT^-EVs, EVs released by VSMCs wildtype; OCR, oxygen consumption rate; PER, proton efflux rate; FCCP, carbonyl cyanide *p*-trifluoromethoxy-phenylhydrazone; AA, Antimycin A; Rot, Rotenone; 2-DG, 2-deoxy-D-glucose; SEM, standard error of the mean.

**Figure 6 ijms-23-13065-f006:**
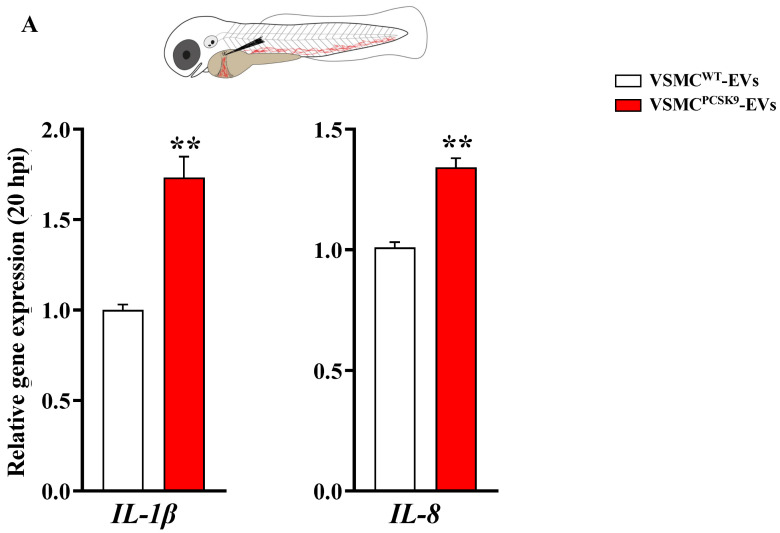
Systemic and local microinjection of extracellular (EVs) in embryos of zebrafish. (**A**) Gene expression of IL-1β and IL-8 evaluated 20 hpi of EVs in embryos. Data were expressed as mean ± SEM of at least two independent biological replicates; for IL-1β and IL-8 genes, statistical significance was assessed by one-way ANOVA test followed by Tukey’s post hoc correction. (**B**) Immuno-detection assay (anti-l-plastin) in brain ventricle area of embryos treated with EVs. A scheme of injection strategy is reported. Representative fluorescence overlaid with brightfield images shows macrophage recruitment (red) at injection site (dashed line) 6 h post-injection. (**C**) Macrophage quantification (l-plastin positive cells); the mean number of cells per embryo ± SEM are indicated (VSMC^WT^-EVs = 22, VSMC^PCSK9^-EVs = 17, from three biological replicates). Red bar represents the injection of VSMC^PCSK9^-EVs; white bar represents the injection with VSMC^WT^-EVs. Statistical significance was assessed by two-tailed unpaired t-test; scale bar 300 μm. * *p* < 0.05, ** *p* < 0.01. EVs, extracellular vesicles; hpi, hours post injection; IL, interleukin; SEM, standard error of the mean; VSMC^PCSK9^-EVs, EVs released by VSMCs overexpressing PCSK9; VSMC^WT^-EVs, EVs released by VSMCs wildtype.

**Table 1 ijms-23-13065-t001:** List of proteins significantly more abundant in vascular smooth muscle cells overexpressing PCSK9 (VSMCs^PCSK9^).

Accession	Description	Peptide Count	Unique Peptides	Confidence Score	*p* Value	Max Fold Change
P20592; P20591	Interferon-induced GTP-binding protein Mx2	2	2	10.92	2.88 × 10^−5^	3.57
P13807	Glycogen (starch) synthase_ muscle	4	4	23.71	1.83 × 10^−13^	2.95
P68036; A0A1B0GUS4	Ubiquitin-conjugating enzyme E2 L3	5	5	53.67	1.97 × 10^−13^	2.72
Q02750; P36507	Dual specificity mitogen-activated protein kinase kinase 1	6	2	36.97	4.70 × 10^−7^	2.64
P17302	Gap junction alpha-1 protein	2	2	13.03	9.33 × 10^−8^	2.36
P04075; P05062	Fructose-bisphosphate aldolase A	15	7	154.95	1.12 × 10^−13^	2.19
Q13228	Methanethiol oxidase	3	3	17.18	3.04 × 10^−8^	2.07
Q96HE7	ERO1-like protein alpha	6	3	37.10	1.39 × 10^−10^	2.03
P00338	L-lactate dehydrogenase A chain	17	10	149.76	2.92 × 10^−12^	2.01
P00558	Phosphoglycerate kinase 1	17	11	169.71	3.76 × 10^−11^	1.87
P30044	Peroxiredoxin-5_ mitochondrial	4	4	39.60	2.53 × 10^−8^	1.84
Q9Y6B6	GTP-binding protein SAR1b	4	2	34.89	0.000135	1.81
O00499	Myc box-dependent-interacting protein 1	4	3	20.62	1.42 × 10^−9^	1.75
Q9BVA1	Tubulin beta-2B chain	20	3	179.13	0.001302	1.74
Q12797	Aspartyl/asparaginyl beta-hydroxylase	4	3	19.84	2.91 × 10^−11^	1.71
Q10471	Polypeptide *N*-acetylgalactosaminyltransferase 2	5	2	27.74	6.82 × 10^−8^	1.69
Q12840	Kinesin heavy chain isoform 5A	14	4	86.31	4.71 × 10^−7^	1.66
Q8NBP7	Proprotein convertase subtilisin/kexin type 9	6	5	39.04	2.34 × 10^−6^	1.65
P43490	Nicotinamide phosphoribosyltransferase	7	4	40.76	1.29 × 10^−8^	1.64
P23381	Tryptophan-tRNA ligase_ cytoplasmic	5	4	33.11	3.90 × 10^−10^	1.64
P48059	LIM and senescent cell antigen-like-containing domain protein 1	2	2	11.07	4.80 × 10^−5^	1.63
Q15050	Ribosome biogenesis regulatory protein homolog	2	2	11.53	2.63 × 10^−6^	1.61
O95379	Tumour necrosis factor alpha-induced protein 8	2	2	11.95	0.000177	1.60
P07948	Tyrosine-protein kinase Lyn	3	2	19.58	7.27 × 10^−9^	1.58
P31150	Rab GDP dissociation inhibitor alpha	12	4	99.46	1.58 × 10^−7^	1.57
P14854	Cytochrome c oxidase subunit 6B1	3	3	17.72	1.91 × 10^−13^	1.56
Q06330	Recombining binding protein suppressor of hairless	5	4	27.02	1.45 × 10^−8^	1.56
P60174	Triosephosphate isomerase	12	7	99.95	2.06 × 10^−8^	1.55
Q9HC10	Otoferlin	4	2	20.14	3.29 × 10^−8^	1.54
P68366	Tubulin alpha-4A chain	18	2	187.63	3.90 × 10^−8^	1.53
P46108	Adapter molecule crk	4	4	25.52	3.67 × 10^−6^	1.53
P58546	Myotrophin	3	2	18.91	4.77 × 10^−5^	1.52
Q9BQE3	Tubulin alpha-1C chain	22	2	232.79	5.32 × 10^−7^	1.50

**Table 2 ijms-23-13065-t002:** List of proteins significantly more abundant in extracellular vesicles released from vascular smooth muscle cells overexpressing PCSK9 (VSMC^PCSK9^-EVs).

Accession	Description	Peptide Count	Unique Peptides	Confidence Score	*p* Value	Max Fold Change
Q8NBP7	Proprotein convertase subtilisin/kexin type 9	12	12	105.04	8.05 × 10^−5^	7.10
O95967	EGF-containing fibulin-like extracellular matrix protein 2	7	7	64.03	2.27 × 10^−4^	4.30
Q9Y6C2	EMILIN-1	12	12	92.98	8.26 × 10^−5^	3.42
P04908	Histone H2A type 1-B/E	3	3	23.21	1.21 × 10^−3^	2.90
P62805	Histone H4	5	5	38.37	3.81 × 10^−4^	2.76
P35442	Thrombospondin-2	10	9	61.02	4.07 × 10^−4^	2.61
P01024	Complement C3	7	7	44.15	3.35 × 10^−4^	2.40
Q14112	Nidogen-2	10	9	64.07	1.09 × 10^−5^	2.29
P17987	T-complex protein 1 subunit alpha	3	3	16.69	4.26 × 10^−3^	2.25
Q9UQP3	Tenascin-N	4	4	22.53	2.15 × 10^−3^	2.08
P98160	Basement membrane-specific heparan sulfate proteoglycan core protein	40	40	264.35	2.40 × 10^−4^	2.01
O60814	Histone H2B type 1-K	5	5	36.96	5.82 × 10^−3^	1.83
Q8WUM4	Programmed cell death 6-interacting protein	15	15	97.41	2.69 × 10^−3^	1.55
P78509	Reelin	7	7	37.44	8.13 × 10^−3^	1.55

## Data Availability

Data will be available upon request to the corresponding author.

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
