# Peer review of "PCSK9 Confers Inflammatory Properties to Extracellular Vesicles Released by Vascular Smooth Muscle Cells"

_ijms, 2022, doi:10.3390/ijms232113065_

Round 1

Reviewer 1 Report

The present study demonstrates the pro-inflammatory role of PCSK9 by influencing the cargo of extracellular vesicles released by VSMC. The manuscript is well structured as well as the experimental design. I have just few concerns about some points:

1. In the first part of the manuscript authors aimed to characterize VSMCPCSK9-EV by using different techniques. They demonstrated EV-derived B1-integrin, alix and PCSK9 amount by western blot analysis. How did the authors normalize the protein expressions?

2. The authors evaluated the EV-induced migration of monocytes and macrophages. In line 268, they pointed out technical issues encountered with THP1 (human monocytic cell line) differentiation in derived-THP1-macrophages. To overcome this problem, they used J774 macrophages (murine macrophagic cell line). Why the authors did not consider either human cell line (eg. U937) or to differentiate peripheral blood monocyte into MDM?

3. The authors assessed the uptake of oxLDL-derived cholesterol by macrophages derived from THP1 to evaluate the contribute of VSMCPCSK9-derived extracellular vesicles on atherosclerotic process. They found an increased oxLDL uptake in THP1-derived macrophages probably due to an upregulation of the scavenger receptor CD36. Did the authors try to inhibit the CD36 in order to confirm this hypothesis?

4. minor concerns:

- Figure legend 1E: please specify the technique used to evaluate the morphology and, if possible, the magnification used;

- Figure 2A: please add to the axes of the graph the annexin the the PI respectively;

- Figure 5C: on the y-axis it would be better to use “cholesterol uptake” instead of “cholesterol”

- Line 407: please correct the typing error related to IL1.

Author Response

The present study demonstrates the pro-inflammatory role of PCSK9 by influencing the cargo of extracellular vesicles released by VSMC. The manuscript is well structured as well as the experimental design. I have just few concerns about some points:

We thank the reviewer for the thoughtful comments

  1. In the first part of the manuscript authors aimed to characterize VSMCPCSK9-EV by using different techniques. They demonstrated EV-derived B1-integrin, alix and PCSK9 amount by western blot analysis. How did the authors normalize the protein expressions?

We have followed the position statement of the International Society for Extracellular Vesicles (Théry C et al., J Extracell Vesicles. 2018 Nov 23;7(1):1535750. doi: 10.1080/20013078.2018). The expression of these proteins is qualitative, since CD63, CD9, Alix and Beta integrin are already constitutive proteins express on the outer surface of extracellular vesicles.

  1. The authors evaluated the EV-induced migration of monocytes and macrophages. In line 268, they pointed out technical issues encountered with THP1 (human monocytic cell line) differentiation in derived-THP1-macrophages. To overcome this problem, they used J774 macrophages (murine macrophagic cell line). Why the authors did not consider either human cell line (eg. U937) or to differentiate peripheral blood monocyte into MDM?

We thank the reviewer for this comment. We agree that using U937 human cells would have been more appropriate, but since our co-authors (F. Zimetti and MP Adorni) are best accustomed to working with these cells we took advantage of this strain (J774) to improve the quality of experimental work. However, in the limitation section we reported the following sentence “Third, instead of using murine J774 macrophages, the human U937 could have been used. However, these two cell lines are commonly used to evaluate phagocytosis, cytokine production, and morphological analysis [83].”

Further, in our country an Ethics Committee authorization is required in the case of peripheral blood monocytes, this aspect has

  1. The authors assessed the uptake of oxLDL-derived cholesterol by macrophages derived from THP1 to evaluate the contribute of VSMCPCSK9-derived extracellular vesicles on atherosclerotic process. They found an increased oxLDL uptake in THP1-derived macrophages probably due to an upregulation of the scavenger receptor CD36. Did the authors try to inhibit the CD36 in order to confirm this hypothesis?

We agree completely with the reviewer that silencing CD36 would have disentangled the role played by this receptor on the uptake of oxidized LDL. We did not test this hypothesis. If mandatory, we need to request three months of extension for the revision. In the meantime, this point has been listed among the limitations as follows “Fourth, to disentangle the role of CD36 in the uptake of oxLDL, an experiment in which CD36 was silenced should have been performed”

  1. minor concerns:

- Figure legend 1E: please specify the technique used to evaluate the morphology and, if possible, the magnification used;

The legend now reads as follows “VSMCs were cultured in standard conditions for seven days and morphology was evaluated by optical microscopy [17]”.

- Figure 2A: please add to the axes of the graph the annexin the PI respectively;

Thank you, this info has been added.

- Figure 5C: on the y-axis it would be better to use “cholesterol uptake” instead of “cholesterol”

Thank you for this suggestion. Cholesterol uptake has been used.

- Line 407: please correct the typing error related to IL1.

Apologies, typing mistakes have been corrected throughout the manuscript.

Reviewer 2 Report

Please present table 1 and 2 in a plot manner. What is the difference between these two tables? What are the samples used for proteomic study? 

The authors did several staining for the cells, such like PKH67 and FITC annexin V. Please not only provide the quantification, but also provide the staining images. 

Author Response

Please present table 1 and 2 in a plot manner. What is the difference between these two tables? What are the samples used for proteomic study?

Apologies for the mistake pertaining to titles of table 1 and table 2. It has been amended.

Table 1 now reads as follows “List of proteins significantly more abundant in vascular smooth muscle cells overexpressing PCSK9 (VSMCsPCSK9)”

Table 2 now reads as follows “List of proteins significantly more abundant in extracellular vesicles isolated from vascular smooth muscle cells overexpressing PCSK9 (VSMCPCSK9-EVs)”

Overall, samples that underwent proteomic analysis were: vascular smooth muscle cells; extracellular vesicles released from vascular smooth muscle cells; secretome of THP-1 exposed to EVs.

Volcano plots have been added in supplemental materials (Supplemental Figure S1 and S3).

The authors did several staining for the cells, such like PKH67 and FITC annexin V. Please not only provide the quantification, but also provide the staining images.

The dot plot of Figure 4A has been added in supplemental materials

Reviewer 3 Report

In this manuscript, Maria Francesca Greco et al. describe in a very elegant way the influences of PCSK9 in the composition of EVs (proteomic and miRNAs) released from human VSMCs and how these EVs impact the cell-to-cell communication among components of atheromatous lesion.

All the methodology and the evidence shown here are very interesting and relevant. There is a minor point correction

1.- Fig 4 A the letters inside the box are not readable

Author Response

In this manuscript, Maria Francesca Greco et al. describe in a very elegant way the influences of PCSK9 in the composition of EVs (proteomic and miRNAs) released from human VSMCs and how these EVs impact the cell-to-cell communication among components of atheromatous lesion.

All the methodology and the evidence shown here are very interesting and relevant. There is a minor point correction

We thank the reviewer for the thoughtful comment.

1.- Fig 4 A the letters inside the box are not readable

Apologies for this flaw. All the letters in figure 4 have been magnified.